# The Effect of Probiotics in a Milk Replacer on Leukocyte Differential Counts, Phenotype, and Function in Neonatal Dairy Calves

**DOI:** 10.3390/microorganisms11112620

**Published:** 2023-10-24

**Authors:** Susan D. Eicher, Janice E. Kritchevsky, Keith A. Bryan, Carol G. Chitko-McKown

**Affiliations:** 1United States Department of Agriculture-Agricultural Research Service, Livestock Behavior Research Unit, West Lafayette, IN 47907, USA; 2Department of Veterinary Clinical Sciences, College of Veterinary Medicine, Purdue University, West Lafayette, IN 47907, USA; sojkaje@purdue.edu; 3Chr. Hansen, Inc., Milwaukee, WI 53214, USA; uskebr@chr-hansen.com; 4United States Department of Agriculture-Agricultural Research Service, Roman L. Hruska Meat Animal Research Center, Clay Center, NE 68933, USA

**Keywords:** probiotic, leukocyte, dairy calves, bovine, respiratory

## Abstract

Probiotics have been investigated for many health benefits; however, few studies have been performed to determine the effects of oral probiotics on peripheral blood and respiratory immune cells in cattle. Our objectives were to determine changes in health and growth status, differential blood cell counts and function, and blood and lung cell function using flow cytometry and PCR in dairy calves fed a milk replacer with (PRO, N = 10) or without (CON, N = 10) the addition of probiotics to the milk replacer and dry rations from birth to weaning. Performance and clinical scores were not different between the treatment groups. Treatment-by-day interactions for peripheral blood leukocyte populations differed in cell number and percentages. A greater percentage of leukocytes expressed the cell surface markers CD3, CD4, CD8, CD11b, and CD205 on d 21 in CON animals. Lung lavages were performed on five animals from each treatment group on d 52. There were no differences between treatment groups for the expression of cytokines and Toll-Like Receptors as measured using Polymerase Chain Reaction, possibly due to the small sample size. Oral probiotics appear to affect peripheral blood immune cells and function. Their effect on overall calf health remains to be determined.

## 1. Introduction

Probiotics are live microorganisms that are taken to improve health by contributing to the gut microbiome. Some foods, such as yogurt and sauerkraut, contain live probiotic bacterial cultures, and other times probiotics are taken orally, separate from other foodstuffs. These supplements often contain prebiotics, which are high in fiber and provide nutrition to the probiotic bacteria. When prebiotics and probiotics are provided in one treatment, they may be called synbiotics since they work in concert with each other to provide a healthy outcome, normally in the gut. Due to increasing concerns about the possible overuse of antibiotics in animal production as well as the possible consequence of pathogens developing antibiotic resistance, probiotics have been under study in animal feeding operations to determine if they can be used as alternatives to antibiotics to decrease disease and to stimulate immunity [1]. Most studies have focused on gut health and the reduction of pathogenic bacteria shedding in feces [1].

Recently, it has been shown that due to the connection between the gut microbiome and other organ systems in the body through the common mucosal immune system, a healthy microbiome in one organ system, such as the gut, can affect the health of other systems [2]. For instance, the microbiome–gut–brain axis, the microbiome–gut–lung axis, and the microbiome–gut–reproductive axis [3]. Little research has investigated the effects of oral probiotics on the respiratory immune cells [4]. Lungs were previously believed to be a sterile environment, but that has recently been revealed not to be the case in the upper respiratory tract [5]. Because an oral route applies to most probiotics, we hypothesized that some of the probiotics may be impacting the respiratory system and the intended enteric system [6,7]. The objectives of this research were to determine changes after the oral feeding of probiotics from birth through weaning in dairy calves resulted in changes to (1) health and growth status, (2) differential cell counts and percentages of peripheral blood, and (3) cell function indicators using flow cytometry and polymerase chain reaction (PCR).

## 2. Materials and Methods

### 2.1. Animals

Animal care and use were approved by the Purdue Animal Care and Use Committee (#1803001701). Twenty calves were placed on the study after meeting the criteria of birth weight between 32 and 50 kg and having a plasma protein value equal to or greater than 5.5 g/dL measured using Brix refractometry between 24 and 48 h after birth. Calves were given 1 L of superior-grade colostrum within 12 h of birth and again within 24 h. Calves were fed 2 L of a 24/20 milk replacer (Milk Specialties Global, Eden Prairie, MN, USA), divided into two equal feedings per day. Beginning on d 2 of life, calves were moved to individual hutches and assigned to control (CON) or probiotic (PRO) treatments of a milk replacer. Probiotics (Bovamine Dairy, Chr. Hansen, Inc., Milwaukee, WI, USA) were delivered into each bottle (2.5 g/bottle) at feeding. Probiotics were kept refrigerated after being aliquoted for each feeding until used. The probiotic consisted of lactose, sodium silico aluminate, dried Propionibacterium freudenreichii fermentation product, and dried Lactobacillus animalis fermentation product (1.5 × 10^9^ CFU/g). Calves were weaned (step-down) at d 42. For example, one milk feeding was discontinued at d 42, and the 2nd was discontinued based on dry feed consumption (after feed consumption at approximately 1.5 kg/d). Probiotics were also included in the dry feed at the recommended dosage (a total of 5 g/day) from d 7 until after weaning was completed. Dry feed was available from d 7 to 52. Calves were weighed at birth and then weekly until d 49. Calves were assigned to a Tuesday or Friday weekly weigh day (whichever was closest to their birth date). Calves were scored daily for fecal scores [8], ocular and nasal discharge, ear orientation, and overall clinical score.

### 2.2. Blood Collection

Jugular blood was collected into 5 mL ethylenediaminetetraacetic acid (EDTA) and 10 mL ACD (acid citrate dextrose) tubes on weigh days 7, 21, 42, and 49. Blood was kept on ice until returned to the laboratory for processing. Blood from the EDTA collection was analyzed for leukocyte differential counts and hematocrit (Genesis, Oxford Science, Oxford, CT, USA). The blood collected into the ACD tubes was processed via hypotonic lysis to isolate the total leukocyte cells [9]. Cells were resuspended into 7 mL of a Roswell Park Memorial Institute 1640 medium (GIBCO, Thermo Fisher Scientific, Waltham, MA, USA). Five hundred microliters of that suspension were aliquoted into four flow cytometry tubes (Falcon 12 × 15 cm, Thermo Fisher Scientific, Carlsbad, CA, USA) for flow cytometry. The remaining suspension was centrifuged at 3000× *g* for 10 min, and the cell pellet was used for RNA extraction using RNEasy mini kits with QIAshredders for cell disruption (Qiagen Sciences, Germantown, MD, USA). The purity of RNA was determined by 260:280, which was calculated on a NanoDrop spectrophotometer. Ratio values of 1.8–2.1 were obtained for the RNA samples used for gene expression analysis.

### 2.3. Lung Lavage

On d 52, lung lavages were performed on five calves from each treatment to obtain bronchial alveolar cells. Cetacaine was sprayed in the left nostril of a calf after moving them to the barn, where they were usually weighed. Calves were gently restrained by two people. The end of a bovine bronchoscope was sprayed with cetacaine and inserted into the trachea. A flexible 10 French catheter (36″) was inserted through the bronchoscope, and 120 mL of sterile saline at 37 °C was infused into the lungs using 60 mL syringes. Immediately after the 120 mL infusion, negative pressure was applied to aspirate the fluid back through the catheter and into a sterile 50 mL endotoxin-free centrifuge tube. The process was repeated to obtain a second sample, if necessary, to obtain 50 mL of lavage fluid. Samples were placed on ice to move to the laboratory. These samples were used for cell phenotyping, activity testing, and Reverse Transcriptase (RT)-PCR.

### 2.4. Flow Cytometry

Peripheral blood leukocytes were labeled with CD3, CD4, CD8, CD14, and CD205 to identify the cell type, and activity was measured with assays measuring CD11b. The phagocytosis and oxidative burst were measured with opsonized *E. coli* bioparticles (Table 1). Briefly, three microliters of antibodies were added to each of the two tubes, and 3 uL of opsonized *E. coli* bioparticles were added to the third tube. The last tube was used as a cell-only control. All were incubated in a 37 °C water bath for 30 min. The CD18-labeled cells were washed one time, and a secondary was added and incubated 30 min more. All other cells were washed with Hank’s Balanced Salt Solution (HBSS) two times before being resuspended in 2% paraformaldehyde in HBSS. Samples were analyzed on a BD Fortessa LSR (BD Biosciences, San Jose, CA, USA) at the Purdue Bindley Bioscience Center flow cytometry lab.

### 2.5. Gene Expression Analysis

The extracted RNA was reverse transcribed using TaqMan RNA reverse transcription kits (Life Technologies, Thermo Fisher Scientific, Carlsbad, CA, USA). The resulting cDNA was then used in TaqMan assays to determine gene expression [10] of a toll-like receptor (TLR) 2, 4, and 9, and IL-6, IL-10, and IL-17.

### 2.6. Statistics

The mixed models procedure in SAS was used with fixed effects of day and treatment and their interaction. Data were tested for normality, and log 10 was transformed as needed. Compound symmetry was determined to be the most appropriate covariance structure. The general linear model (GLM) was used to determine differences in days with fecal scores ≥ 2.5 and lung lavage PCR.

## 3. Results

### 3.1. Performance

Calves were equal in birthweight (90.7 and 91.9 for CNT and PRO, respectively; *p* = 0.44). Additionally, weekly weights and gains did not differ between the CNT and PRO calves (Figure 1). Days with fecal scores greater than or equal to 2.5 were not different between the treatments (7 and 6.3 for CON and PRO, respectively; *p* = 0.72). The ear, ocular, nasal, and clinical scores were not different between the treatments (*p* ≥ 0.10). Overall, adding probiotics to a milk replacer did not improve production in the seven weeks of our study.

### 3.2. Hematology

Total leukocyte counts, neutrophil counts, lymphocyte counts, monocyte counts, and eosinophil counts were not different between CON and PRO treatments (Figure 2). No day effects were detected, however, and treatment-by-day effects were found.

Percentages of lymphocytes and eosinophils did not differ between control and probiotic animals; however, neutrophil, monocyte, and basophil percentages differed by treatment-by-day interactions with the probiotic-treated calves having a higher percentage of neutrophils, and eosinophils on d 21, and a higher percentage of basophils on days 21 and 42 (Figure 3). Monocyte percentages were decreased in probiotic-treated calves on days 21 and 49 (Figure 3).

### 3.3. Flow Cytometry

Lymphocyte phenotypes were assessed using CD3, CD4, and CD8 expressions. No differences were seen for mean fluorescence intensity (MFI) for CD3, C4, or CD8. However, a greater percentage of cells fluorescing was observed in CON animals for each cell surface marker on d 21 (*p* < 0.05; Figure 4).

Antigen-presenting cells were identified via staining for CD205 for dendritic cell (DC) populations, CD14 for monocyte populations, and lipopolysaccharide (LPS) recognition. Neither were different in MFI throughout the study. However, the percent of cells expressing CD205 tended to be greater for CON animals on d 21 (*p* = 0.08). The percent of CD14 CON and PRO was not different (*p* = 0.11) on d 7 and 21 (Figure 5).

The function of cells was measured using adhesion with CD11b serving as the marker, and phagocytosis and oxidative burst were measured with opsonized *E. coli* bioparticles. No differences in MFI were detected (*p* ≥ 0.10); however, the percentage of cells expressing CD11b was greater for CNT at d 21 (*p* < 0.05) (Figure 6). Although the percentage of cells that phagocytositized *E. coli* and had oxidative burst increased by d 21, and no trt effect (*p* ≥ 0.10) or trt * d interactions (*p* ≥ 0.10) were observed (Figure 6).

### 3.4. qRT-PCR Analysis of Gene Expression

TLR4, TLR10, and Interleukin (IL)-10 expression was measured in peripheral blood leukocytes over time. Trends for TLR9 and IL-10 differed over time but were not significant (Figure 7). Expression of TLR 2, 4, and 9 and the cytokines IL-6, IL-10, and IL-17 by leukocytes obtained via lung lavage on d 52 did not differ between treatments (Figure 8).

## 4. Discussion

Although there is increasing literature on probiotics and the intestinal microbiome, there are few studies on the effects of probiotics on respiratory immunity in cattle, and studies on calves receiving a milk replacer with probiotics are limited. We delivered probiotics within 24 h of birth, aiming to modulate respiratory health. Loyd and Saglani [11] determined in a review that alterations in the respiratory system early in life can determine lifelong lung health. Lima et al. [12] showed changes in healthy and diseased calves’ upper respiratory tract microbiomes from 3 days of age to 35 days of age. Bosch et al. [13] suggested that imbalances of the upper respiratory tract microbiome may lead to invasion and overgrowth by pathogenic bacteria. Holman et al. [4] determined that the microbiome of cattle on the day of arrival into a feedlot and after 60 were significantly different. However, Corbett et al. [14] noted that feeding probiotics did not reduce susceptibility to respiratory disease in cattle. Additionally, Adjei-Fremeh et al. [6] reported that feeding of probiotics induced global gene expression upregulation of genes associated with both innate and adaptive immunity; cytokine and chemokines, TLRs, and stress-related signaling molecules that are related to the inflammatory response and the maintenance of homeostasis were predominant. Furthermore, a recent review stated that Lactobacillus and Lactococcus are associated with good respiratory health and that internasal delivery modifies the nasopharyngeal microbiota and can protect against opportunistic pathogens [15]. We used an oral route, which can often result in nasal and oral delivery when calves eat dry feed or drink milk.

In our study, calves received commercial probiotics in the milk replacer after birth, followed by the addition of probiotics to the dry ration during step-down weaning beginning at day 42. Our trial ended on day 52. No differences were noted between the CON and TRT groups for any of the production or health parameters measured. Similar to our study, others have found no differences in average daily gain in calves fed probiotics compared to controls and other treatments [16,17,18]. Of note are studies comparing production traits between animals on a control diet or a diet containing Monenesin or probiotics [19,20]. The animals on the probiotic diet gained as much as those on the diet containing Monensin. A separate group of studies shows the opposite effect—the inclusion of probiotic bacteria or yeast products in diets results in increased average daily gain and improved production [21,22,23]. It should be noted that none of the calves in our study had diarrhea, so it is not possible to determine if the addition of probiotics aided in lessening the severity or time until clearance, as reported by others [16,18,24,25,26,27,28,29].

When peripheral blood parameters were measured, we found that neutrophil, monocyte, and basophil percentages differed in treatment-by-day interactions. These differences were also seen when comparing the total cell population counts. These differences may indicate that the effects of probiotics on immune cell populations can vary over time and are not static.

Lymphocyte populations were examined using flow cytometry based on the expression of CD3 to identify all T cells, CD4 to identify T helper cells, and CD8 to identify cytotoxic T cells important in adaptive immunity. Although no differences were seen for MFI, a greater percentage of cells expressing all three markers at d21 in the CON animals were identified. Dairy cattle fed probiotic bacteria at the onset of mastitis had decreased days on medication and increased CD4+ and CD11c, Cd172high dendritic cells [30]. Because none of the animals in our study showed signs of illness, this may indicate that the probiotics dampened the immune response of the treated calves when it was not needed but would ramp up immunity if infection was detected. In this study, we used CD14 as part of the LPS recognition molecule because recognition of Gram-negative bacteria requires the expression of CD14. CD18 was included as a marker of cell activation and adhesion, and CD205 was used to determine the role of dendritic cells in the phagocytosis of *E. coli* bioparticles, and the associated oxidative burst to determine phagocytic function. Similar to the lymphocyte results, we saw a decrease in the percentage of cells expressing CD11b and CD205 at d21 in the PRO group. As stated above, if an infection is not present, a dampening of the immune response is an efficient response.

In peripheral blood cells, we measured the expression of TLR4 since it is a pattern-recognition receptor (PRR) for LPS, TLR9 a PRR for CpG motifs found in viruses and bacteria, and IL-10 which plays a role in limiting the immune response. These factors are mainly produced by the monocyte population. Although the expression appears to vary over time in both CNT and PRO animals, they were not statistically different; thus, the addition of probiotics to the diet did not appear to alter the ability of calves to respond to infection with either viruses or bacteria nor did their ability to control the immune response by IL-10 differ. Stimulation of bovine PBMC with *Lactiplantibacillus plantarum* showed increased expression of IL-1β, IL-6, and IL-10 from CD14+ cells through TLR 2/4 signaling [31].

When we measured lung lavage cells for the expression of TLR2 a PRR that recognizes some protozoa as well as TLR4, TLR9, and cytokines IL-6, IL-10, and IL17, we found no differences in expression between the two treatment groups. Thus, the addition of probiotics to the diet did not result in changes in the expression of immune mediators in the lung compartment either due to accidental inhalation of the probiotics during eating or through the common mucosal immune system.

Dairy cows supplemented with *Sacchromyces cerevisiae* had higher levels of IL-1β in plasma, greater milk yield, and reduced risk of metabolic disease [32]. Exposure of human osteoblast and lung epithelial cell lines treated with β-glucan had increased production of IL-6 and IL-8 in response to *Staphylococcus areus* [33]. Probiotics also modulate the response of the mammary gland to pathogen challenge, resulting in the expression of inflammatory cytokines through regulation of the TLR signaling pathway [34]. *L. plantarum* decreased the expression of TLR2, TLR4, IL-1β, IL-6, and IL-8, among other factors in bovine mammary epithelial cells exposed to *E. coli,* thereby reducing inflammation [35]. Bovine intestinal epithelial cells treated with Lactobacillus mucosea showed modulation of TLR3-mediated immunity and TLR4-mediated inflammation [36].

Most literature reports on *Lactobacillus* strains in disease prevention of pneumococcal infections [37] and *Lactobacillus* have been used to determine some mechanisms that reduce susceptibility in vitro [38]. However, *Bacillus subtilis* delivered intranasally increased TLR expression in the tonsils of pigs [39]. Monocyte-derived DCs were not affected in numbers or maturation by the soluble mediators of *Lactobacillus rhamnosus*, but their capacity to modulate T cell responses was enhanced [40]. Additionally, *Lactobacillus rhamnosus* CLR 1505 modulated the TLR3-mediated immune response in the respiratory tract of mice [41]. Lehtoranta et al. [42] reviewed the effectiveness of some common probiotics in humans and mice. They concluded that the variability in outcomes may be attributed to the strains of probiotics in use, bacterial dose, and matrices provided with the probiotics. We found similar results in an in vitro study that exposed lung lavage cells from cattle to different species of probiotic bacteria [43]. TLR3 is important in viral inflammatory responses and pathology. Our data showed an increase in the number of cells expressing the CD205 dendritic cell marker, tending to decrease with oral probiotic delivery in neonatal calves. In concurrence with Forsythe’s and Kunze’s observation that microbes have effects on dendritic cell phenotype and function [44], our data show that dendritic cells certainly play a role in the ability of the leukocytes to modulate immunity. Our data also show decreased activation. The increase in the percentage of cells with oxidative burst corresponds with the increase in the percentage of cells expressing the DC marker. This would be a desirable characteristic of a probiotic affecting the respiratory tract.

Nasally delivered *L. lactis* NZ900 improved clearance of *S. pneumoniae*, possibly using a competitive exclusion mechanism [37] and by enhanced IgA and IgG in BAL fluid in mice. Marranzino et al. demonstrated that TNF-α concentrations were not altered in BAL compared with serum and intestinal fluid, but IFN-γ was increased by two or three strains of *Lactobacillus* compared to controls in BAL, both in ex vivo and in vitro experiments [45].

The oxidative burst of those two strains was also greater than that of controls [45]. Cell counts of pathogenic *C. albicans* in the lungs of infected mice showed a reduction with *L. casei* CRL431 and *L. rhamnosus* CRL1505 treatments. There are numerous differences in the approaches used in these two studies. Marranzino et al. [45] did much of their study in vivo in mice. Method of delivery and duration of supplementation have been cited as reasons for differences in the effectiveness of probiotic supplements on upper respiratory symptoms; some showed benefit in rate while others showed a reduction in duration or severity but not in incidence.

Because we used a static system, in vitro, we would not expect large shifts in cell population percentages, such as in our phagocytosis data, where little change was evident in the percentage of cells, but the mean expression showed some substantial differences. It is possible that effects in vivo may be more dramatic because of the increased chance of affecting the cell population development.

Other benefits attributed to probiotics are increased expression of mucin genes and mucin secretion in intestines [46] and antimicrobial peptide-producing cells; whether that is true for respiratory mucosal surfaces is not known. Additionally, many probiotics have mechanical actions that are antagonistic to pathogens [47].

## 5. Conclusions

Recent reviews report that the timing of probiotic administration, the strain of probiotics used, and the experimental design overall can lead to differences in results [48,49]. We have found this to be true in past studies comparing probiotic species in vitro [43]. In this study, probiotics provided orally in dairy calves in a milk replacer and dry rations appear to affect peripheral blood immune cells and function. Their effect on overall calf health remains to be better determined.

## Figures and Tables

**Figure 1 microorganisms-11-02620-f001:**
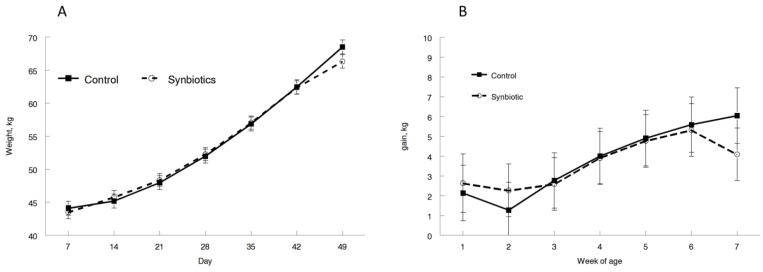
Weight (**A**) and weight gain (**B**) over time of dairy calves fed a milk replacer without probiotics (control) or with probiotics (probiotics).

**Figure 2 microorganisms-11-02620-f002:**
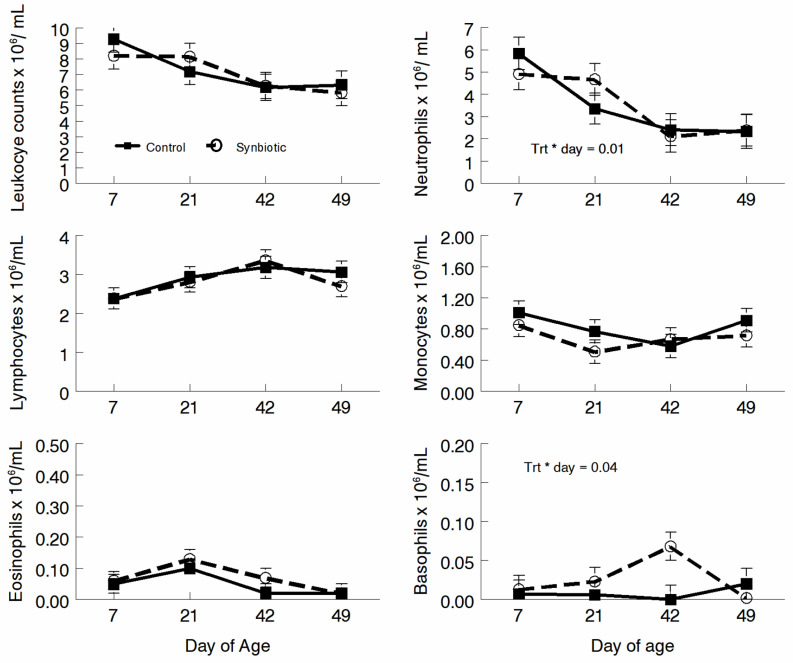
Five-part differential counts of jugular blood samples in EDTA on d 7, 21, 42, and 49 for calves given a milk replacer only (control) or a milk replacer supplemented with probiotics (probiotics).

**Figure 3 microorganisms-11-02620-f003:**
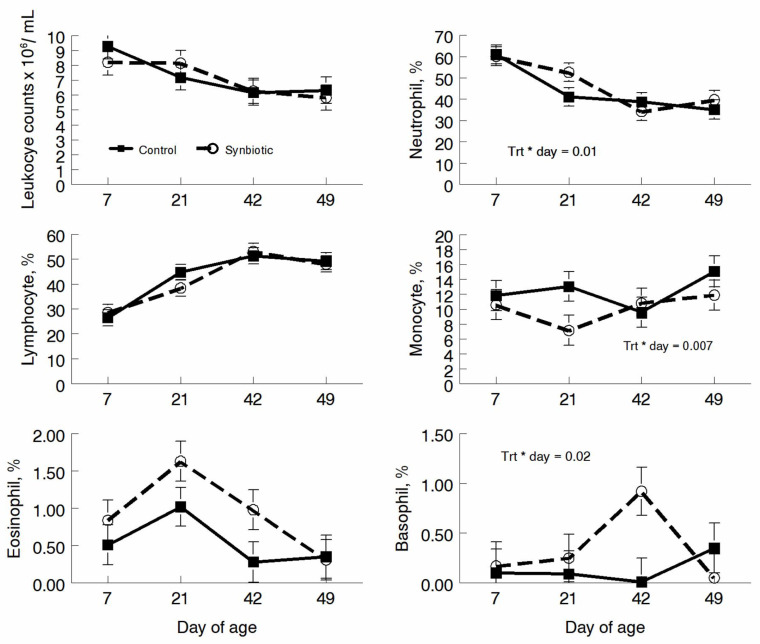
Total leukocyte counts and percentages of cells that were lymphocytes, neutrophils, monocytes, eosinophils, or basophils at 7, 21, 42, or 49 days of age for calves fed a milk replacer alone (control) or a milk replacer supplemented with probiotics (probiotic).

**Figure 4 microorganisms-11-02620-f004:**
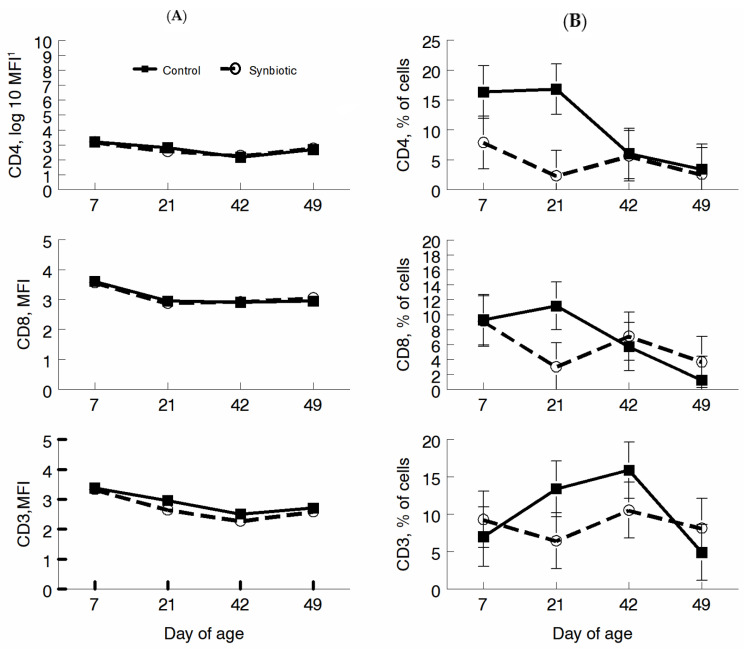
Lymphocyte phenotypes of calves given a milk replacer or a milk replacer with probiotics on d 7, 21, 42, and 49 of age. Mean fluorescence intensity (MFI; (**A**)) and percentage of cells expressing CD3, CD4, or CD8 (**B**).

**Figure 5 microorganisms-11-02620-f005:**
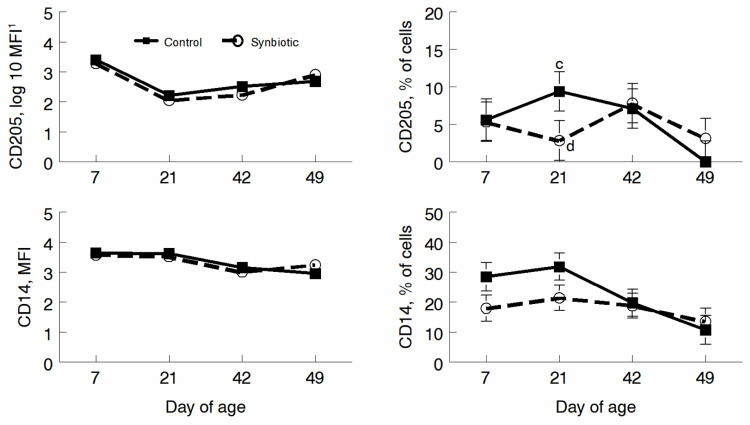
Mean fluorescence intensity (MFI) and percentage of cells fluorescing for antigen-presenting cells using CD14 (monocytes) and CD205 (dendritic cells) on d 7, 21, 42, and 49. Calves were fed a milk replacer only or a milk replacer with a probiotic supplement. Points with differing letters tend towards significance (*p* = 0.08).

**Figure 6 microorganisms-11-02620-f006:**
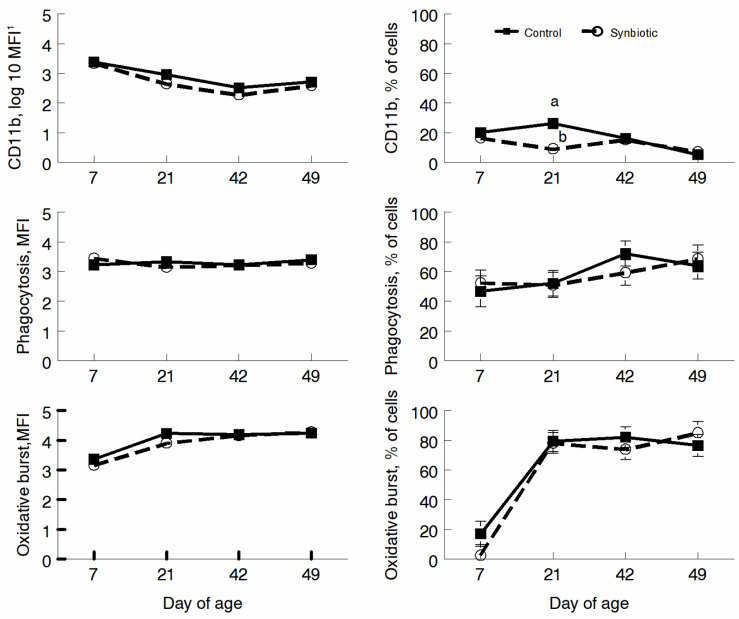
Mean fluorescence intensity (MFI) and percentage of cells fluorescing (%) for Cd11b (activation and adhesion marker), phagocytosis of opsonized *E. coli* bioparticles, and resulting oxidative burst on d 7, 21, 42, and 49. Calves were fed a milk replacer only (control) or a milk replacer supplemented with probiotics (probiotics). Points with differing letters are significantly different (*p* < 0.05).

**Figure 7 microorganisms-11-02620-f007:**
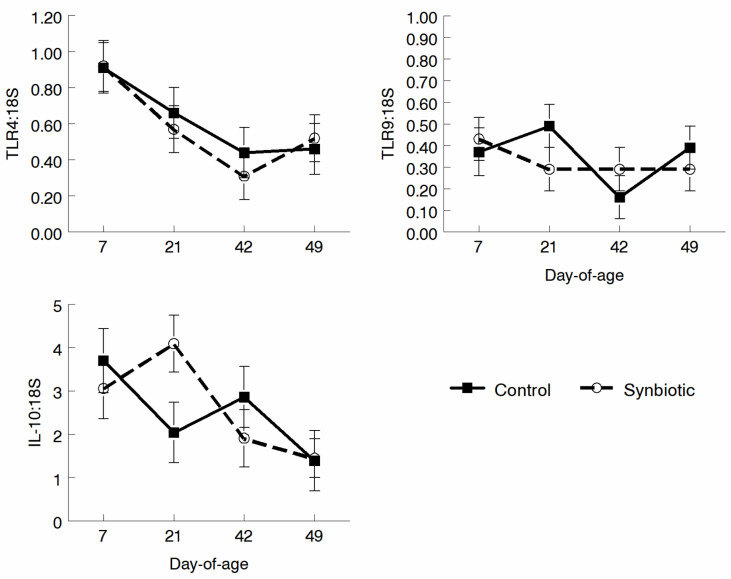
Expression of TLR 4, TLR 9, and IL-10 in peripheral blood leukocytes as measured using qRT-PCR. The expression of TLR 4 and 9 was not statistically different; however, overall trends for TLR 9 varied over time, as did the expression of IL-10.

**Figure 8 microorganisms-11-02620-f008:**
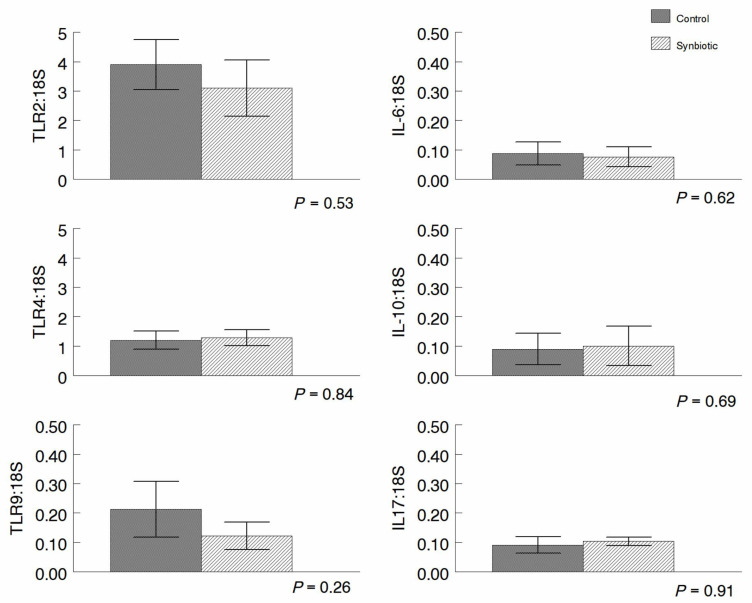
Expression of TLR 2, 4, 9 and IL-16, -10, and 17 in lung lavage cells as measured using quantitative (q)RT-PCR. Expression of the TLR 2, 4, and 9 on the surface of leukocytes obtained by lung lavage does not differ, nor does expression of the anti-inflammatory cytokines IL-6, IL-10, nor the pro-inflammatory cytokine IL-17 by these cells.

**Table 1 microorganisms-11-02620-t001:** Antibodies used in flow cytometry and their supplier.

Antigen (Ab)	Flurorphore	Source
CO	0	---
CD4	FITC	BioRad ^1^
CD8	PE	BioRad ^1^
CD14	RPE	BioRad ^1^
CD18	FITC	WSU ^2^, BioRad goat anti-mouse secondary Ab ^1^
CD205	RPE	BioRad ^1^
CD3	RPE	BioRad ^1^
CD11b	FITC	BioRad ^1^
Phagocytosis	*E. coli* bioparticles pHrodo Red + OPSONIN	Invitrogen ^3^

^1^ BioRad Life Science, Hercules, CA, USA. ^2^ WSU Monoclonal Antibody Center, Pullman, WA, USA. ^3^ Invitrogen, Thermo Fisher Scientific, Waltham, MA, USA.

## Data Availability

The data presented in this study are available upon request.

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
