# Peer review of "The Effect of Probiotics in a Milk Replacer on Leukocyte Differential Counts, Phenotype, and Function in Neonatal Dairy Calves"

_microorganisms, 2023, doi:10.3390/microorganisms11112620_

Round 1
Reviewer 1 Report
If the authors wanted to evaluate the overall benefit of the probiotics an the lungs health, the animals should be challenged with respiratory pathogen. Because no differences were noted between probiotic and normal food.
I would suggest adding the flow cytometry gating strategy as supplementary material.
Author Response
The authors thank the reviewer for their suggestions. We first hoped to determine the mechanisms through which probiotics may effect immunity/protection against infectious respiratory disease prior to performing a pathogen challenge. We intend to naturally expose cattle to respiratory pathogens in subsequent experiments.
Regarding moving the flow cytometry data to the supplemental data section: we would prefer to leave this data in the body of the manuscript but will move them if directed by the editor.
Reviewer 2 Report
Dear Editor, dear authors,
I have read carefully the paper entitled “The effect of probiotics in milk replacer on leukocyte differential counts, phenotype, and function in neonatal dairy calves“.
Article is well written, it is readable, the concept is easy to follow and the scope of work is defined properly. However, there are some issues through the text that need revision. Overall, my suggestion is that the paper should be accepted with minor revisions listed bellow:
1. The introductin part should be expanded. It is not enough only a few sentences about probiotics, the topic should be elaborated more widely.
2. Figures presented in the part Results should be uniformed.
3. The results should be commented in more detail, especially where there are differences in the results between the samples. In the Figure 3, results of basophils are not commented.
Author Response
We thank the reviewer for their comments/suggestions.
(1) The introduction has been expanded, (2) the figures have been adjusted to be more similar - we hope this is what was desired, and (3) we have gone into more detail in the results section. Since this was a "results" and not a "results and discussion" section, we initially limited the information in the first draft.